# The Next Frontier in Neuroprosthetics: Integration of Biomimetic Somatosensory Feedback

**DOI:** 10.3390/biomimetics10030130

**Published:** 2025-02-21

**Authors:** Yucheng Tian, Giacomo Valle, Paul S. Cederna, Stephen W. P. Kemp

**Affiliations:** 1Department of Biomedical Engineering, University of Michigan, Ann Arbor, MI 48109, USA; jyctian@umich.edu (Y.T.); cederna@med.umich.edu (P.S.C.); 2Department of Electrical Engineering, Chalmers University of Technology, SE-412 96 Gothenburg, Sweden; valleg@chalmers.se; 3Section of Plastic Surgery, Department of Surgery, University of Michigan, Ann Arbor, MI 48109, USA

**Keywords:** neuroprosthetics, somatosensory feedback, biomimetic, neurostimulation, neuroelectronics, surgical technique

## Abstract

The development of neuroprosthetic limbs—robotic devices designed to restore lost limb functions for individuals with limb loss or impairment—has made significant strides over the past decade, reaching the stage of successful human clinical trials. A current research focus involves providing somatosensory feedback to these devices, which was shown to improve device control performance and embodiment. However, widespread commercialization and clinical adoption of somatosensory neuroprosthetic limbs remain limited. Biomimetic neuroprosthetics, which seeks to resemble the natural sensory processing of tactile information and to deliver biologically relevant inputs to the nervous system, offer a promising path forward. This method could bridge the gap between existing neurotechnology and the future realization of bionic limbs that more closely mimic biological limbs. In this review, we examine the recent key clinical trials that incorporated somatosensory feedback on neuroprosthetic limbs through biomimetic neurostimulation for individuals with missing or paralyzed limbs. Furthermore, we highlight the potential impact of cutting-edge advances in tactile sensing, encoding strategies, neuroelectronic interfaces, and innovative surgical techniques to create a clinically viable human–machine interface that facilitates natural tactile perception and advanced, closed-loop neuroprosthetic control to improve the quality of life of people with sensorimotor impairments.

## 1. Introduction

Limb dexterity is vital, as it enables a wide range of physical interactions with the environment that are essential for daily activities, work, and recreation [1,2]. The loss of limb function significantly affects individuals’ quality of life, impacting mobility, balance, coordination, communication, and independence [3,4,5]. In the United States alone, over 2 million people are living with limb loss due to causes such as trauma, diabetes, and peripheral vascular diseases [6]. Additionally, there are more than 300,000 individuals in the United States with a spinal cord injury (SCI), many of whom suffer from loss of limb function [7,8]. A majority of these individuals face functional and psychosocial limitations due to their disability [9,10]. In recent decades, the development of neuroprosthetics—robotic devices that restore missing limb functions—has made substantial progress, offering people with limb loss and impairment more effective replacements [11,12,13,14,15]. Clinical trials showed that current neuroprosthetic limbs can reliably be used to precisely manipulate and transfer objects and complete complex tasks, such as making coffee and navigating obstacles with multiple degrees of freedom (DoFs) [16,17,18,19,20,21,22,23]. Despite these advancements, the absence of somatosensory feedback has been a major limitation in current devices [24,25,26]. The sense of touch, for instance, plays an important role in object manipulation in everyday tasks [27,28] (such as opening a water bottle), emotional connections [29] (such as affective touch), and social and psychosocial well-being [30,31] (self-esteem and participation in social activities). Without this somatosensory feedback, neuroprosthetic devices may feel burdensome and lack a natural sense of integration [32].

To improve naturalness and functionality, researchers have integrated tactile feedback in limb neuroprosthetics [33,34,35]. Clinical studies showed that electrically stimulating either the peripheral nerves [34,36,37,38,39] or somatosensory cortex [33,35,40] can evoke tactile sensations in individuals with limb loss or impairment (e.g., paralysis). By incorporating somatosensory feedback into the control loop of neuroprosthetics, users experience improved embodiment of the device while the cognitive effort required for its use decreased [11,41,42,43]. For individuals with limb loss, several key clinical studies found that stimulating peripheral nerves previously innervating the limbs—such as the median, ulnar, radial, and tibial nerves—can evoke vivid tactile sensations perceived on the missing limb and described as touch, pressure, tapping, and vibration [37,44,45,46]. For individuals with intact but paralyzed limbs after an SCI or a brachial plexus injury, research has focused on stimulating the primary somatosensory cortex (S1) directly targeting hand representation. This would potentially allow for localized tactile sensations on the hand [33,35,40]. Advances in neurostimulation allowed researchers to modulate the intensity and naturalness of electrically evoked sensation at both the peripheral nervous system (PNS) and central nervous system (CNS) levels [37,40,44,46,47]. By varying stimulation parameters, including the pulse amplitude, width, frequency, and duration, participants reported different qualities of tactile sensations [46,47,48,49,50,51,52]. The ability to fine-tune somatosensory feedback implies that the design of neurostimulation strategies could drive progress in bidirectional (sensorimotor) neuroprosthetic limbs for enhanced, naturalistic control. However, challenges remain: the induced tactile sensations are not yet consistently stable nor are they perceived as naturalistic over time and across subjects. The current sensory stimulation methods to evoke tactile perception are somewhat arbitrary and unnatural. They are quite simple and imprecise and usually do not correspond to how biological touch encodes sensory information [53,54]. This limits the intuitiveness and functionality of sensorimotor neuroprosthetic control, leading to poor device embodiment and increased cognitive load. Achieving a complete and natural somatosensory experience—one that makes users feel as if they have their functional hands again—requires mimicking the biological processes of somatosensory transduction [53]. In biology, these biological sensors detect tactile stimuli applied on the skin and relay the information to the brain for processing and perception [55]. The best way to modulate the nervous system to improve the naturalness of tactile sensation is to communicate using a “biological neural language” that seamlessly integrates motor and sensory systems requiring minimal learning [56,57]. This concept, known as biomimetic somatosensory feedback, has evolved significantly over the past decade and marks a crucial step toward the development of bionic limbs with more natural, intuitive functionality [53,54].

In clinical trials, researchers developed and implemented various biomimetic encoding strategies designed to translate tactile information—captured by tactile sensors embedded in the neuroprosthetic limbs—into patterns of neurostimulation [58]. These methods use mathematical equations and computational techniques to model how sensory neurons respond to tactile stimuli, such as skin motion, deformation, or indentation. This information is then exploited to generate appropriate neurostimulation trains. In biological systems, mechanoreceptors like Meissner corpuscles (for detecting object slippage), Pacinian corpuscles (for vibratory cues during grasping), Merkel cells (for texture and form perception), and Ruffini corpuscles (for sensing skin stretch and object movement) respond to the mechanical deformation of the skin [59]. These mechanosensory neurons are generally categorized into two main types: slowly adapting (SA) neurons, which fire consistently in response to sustained indentation, and rapidly adapting (RA) neurons, which fire rapidly in response to changes in skin pressure [59,60]. Deflorio et al. comprehensively reviewed the current computation models that mimic mechanoreceptor responses to tactile stimuli [61]. Using these neuron models, researchers generate neuron spike trains that inform strategies of electrical neurostimulation. This biomimetic approach provides biologically inspired somatosensory feedback to individuals who have lost or impaired natural somatosensory processing [62].

This review highlights key human clinical studies that utilize in silico neuron models to generate biomimetic somatosensory feedback provided to neuroprosthetic users through neurostimulation. These investigational studies demonstrated encouraging results in effectively encoding tactile stimuli within the nervous system for more intuitive and functional tactile perception. We also discuss how innovations in advanced electronic skins, neural interfaces, and surgically created sensory interfaces could further improve both clinical and functional outcomes in the field of bidirectional neuroprosthetics.

## 2. Biomimetic Somatosensory Feedback for Upper Limbs

Over the past few years, multiple key human trial evaluations marked significant milestones in the development of limb neuroprosthetics by implementing biomimetic encoding strategies to provide relevant somatosensory feedback for upper-limb prosthesis users, both for people with amputations and for people with an SCI [47,63,64,65,66,67]. These achievements are rooted in the reliability of biomimetic models validated in animal studies prior to clinical application [57,68,69,70]. In these animal studies, the successful assessment of implant safety, effectiveness of neurostimulation strategies, and physiological validation of the restored sensory signaling provided confidence, guidance, and a foundation for the recent clinical translation of biomimetic somatosensory feedback. In 2018, Osborn et al. introduced a prosthetic interface that translates tactile information into biologically relevant neural signals (Figure 1A) [63]. In a participant with a transhumeral amputation of the left arm, surface electrodes were placed on the subject’s residual limb to target the median and ulnar nerves. Transcutaneous electrical stimulation (TENS) of these residual nerves was used to evoke tactile perceptions in the phantom hand [63,71]. In the prosthetics context, Osborn et al. modeled the responses of mechanoreceptors, combining the characteristics of SA and RA receptors using the Izhikevich neuron model [72], which includes regular and fast-spiking neurons, respectively [63,73]. The Izhikevich model is simplistic but remains computationally efficient [72]. Biomimetic tactile sensors, designed to emulate the structure and function of skin (e.g., epidermis and dermis), were integrated into the neuroprosthetic fingertips to measure pressure. These measurements were converted into input currents and fed into the neuron model. The model generated receptor-specific spiking patterns in terms of the neuron voltage and timing, representing a biomimetic approach that replicates natural biological processes and signals. These patterns, corresponding to the tactile interaction with objects during prosthesis grasping, were used to modulate the stimulation parameters, such as frequency and pulse width, which tuned the perceived tactile sensation elicited by TENS [63]. With this biomimetic approach, the participant in the study was able to differentiate objects with different curvatures, including sharpness, through the incorporation of nociceptive tactile feedback [63]. To further demonstrate the functional benefits of biomimetic somatosensory feedback, George et al. conducted a study in which an individual with a transradial amputation was implanted with Utah Slanted Electrode Arrays (USEAs) into the residual median and ulnar nerves [64]. The researchers designed and compared two biomimetic sensory encoding algorithms aimed at replicating natural tactile signals. The first algorithm, a first-order model, incorporated both the magnitude and rate of change of the contact force to mimic phasic bursts of neural activity (SA and RA). The second algorithm, a second-order model [57,74], simulated the sensitivity of sensory nerve fiber populations (SA, RA, and Pacinian corpuscles) to skin indentation and its derivatives, including the rate and acceleration [64]. These biomimetic strategies significantly enhanced the performance in object discrimination tasks compared with the non-biomimetic feedback (Figure 1B). Specifically, the first-order model improved the response times by 24% for size discrimination and 44% for compliance discrimination. The second-order model resulted in 56% faster object compliance identification [64]. This study provided compelling evidence that biologically inspired neurostimulation patterns, which capture and recreate the temporal dynamics of natural tactile signals, produce more interpretable sensory percepts and functionally contribute to more intuitive neuroprosthetic control. In a study by Valle et al., the researchers also found that a hybrid biomimetic approach that combined frequency and amplitude modulation of stimulation achieved an optimal balance between naturalness and sensitivity in restored tactile perception (Figure 1C) [65]. This approach was tested on a participant with a transradial amputation who received transverse intrafascicular multichannel electrode (TIME) implants in the median and ulnar nerves. The amplitude modulation encoder modulated the neurostimulation amplitude linearly based on the force measured by the prosthetic hand sensors. This provided highly sensitive force feedback to the user. The frequency modulation encoder, on the other hand, modulated the frequency of the neurostimulation pulses based on the model output of all three fiber types (SA, RA, and Pacinian corpuscles). This allowed for more natural tactile feedback compared with conventional strategies. When these two encoding strategies were combined, improvements in both gross manual dexterity and neuroprosthesis embodiment were observed [65]. These findings highlight the importance of capturing a comprehensive biomimetic representation of skin dynamics and neural activity rather than focusing on replicating individual features of tactile function. Such biomimetic encoding seems to be critical for achieving naturalistic, bidirectional neuroprosthetic control.

The three important clinical studies detailed above focused on investigating biomimetic tactile feedback through the electrical stimulation of peripheral nerves in individuals with limb loss. The spinal cord or primary somatosensory cortex represents another potential target for neuroelectronic interfaces to deliver biomimetic tactile feedback to neuroprosthetic users who suffer from high-level or proximal-limb amputation (e.g., shoulder or hip disarticulation). However, to our best knowledge, no clinical trials have demonstrated that biomimetic stimulation of the spinal cord or somatosensory cortex can enhance tactile perception and bidirectional neuroprosthetic control in individuals with limb loss yet. On the other hand, intracortical microstimulation (ICMS) of the primary somatosensory cortex by Utah arrays is an emerging technique for providing biomimetic somatosensory feedback for individuals with paralyzed and deafferented limbs following SCI [67]. In 2023, Shelchkova et al. implemented a biomimetic sensory encoder that delivered high-amplitude neurostimulation during both onset and offset, while reducing the stimulation amplitude during the sustained phase of object contact [66]. This approach mimics the biological response, characterized by strong neural signals when an object initially contacts or leaves the skin, and the adaptation of sensory neurons during prolonged touch. In this study, ICMS was applied to the hand-representing region of S1 in participants with hand paralysis [66]. The biomimetic stimulation minimized the disruptions in decoder performance due to the ICMS-evoked motor cortex activation and led to fewer failed trials in object transportation compared with non-biomimetic stimulation [66]. In 2024, Greenspon et al. demonstrated that biomimetic ICMS enhanced the sensitivity to changes in stimulation intensity, increased the number of discriminable stimulation intensity levels, and reduced the total charge needed to evoke comparable sensations (Figure 1D) in participants with SCI [47]. Moreover, when combined with multi-electrode stimulation, biomimetic ICMS significantly improved the task performance. In a compliance discrimination task, multi-electrode biomimetic feedback outperformed single-electrode non-biomimetic feedback, with error rates of 7.5% versus 25%, respectively [47]. In another clinical study conducted by the same group, Hobbs et al. further showed that biomimetic ICMS elicited tactile percepts that more closely resembled natural residual percepts, as induced by mechanical indentation on an insensate area of the hand, compared with non-biomimetic stimulation [67].

## 3. Biomimetic Somatosensory Feedback for Lower Limbs

Unlike the restoration of somatosensation in upper-limb neuroprosthetics, which primarily aims to improve fine motor control, such as object manipulation, the restoration of somatosensation in the lower limb focuses on improving postural stability, gait, and walking speed. Nevertheless, biomimetic approaches are effective in addressing both scenarios. In the past decade, several human studies focused on restoring somatosensory feedback in individuals with lower-limb amputations by utilizing sensors embedded in the plantar surface of the prosthetic foot via a sensorized insole [75,76,77,78]. These pressure sensors, strategically positioned on the insole, measure gait dynamics and pressure at contacts points between the foot and the ground that are relevant during walking [77]. This tactile information is crucial for lower-limb neuroprosthesis users, increasing walking speed, reducing cognitive effort, and helping to maintain balance and navigate challenging terrains [75,79,80]. The tactile feedback in lower-limb neuroprosthetics is typically delivered through electrical neurostimulation of the peripheral nerves. However, to date, few human studies have explored the application of biomimetic tactile stimulation in the lower limb. In 2024, Valle et al. presented a groundbreaking clinical study that demonstrated the effectiveness of biomimetic tactile feedback for individuals with lower-limb loss, marking a significant advancement in the field [68]. The researchers used a computational model called FootSim [81], which was designed to replicate the neural activity of sensory afferents in the human foot in response to skin deformations during walking. Electrical neurostimulation patterns were based on the neural spiking patterns generated by the model. In the study, three participants with transfemoral amputations received tibial nerve implants [68]. Compared with non-biomimetic stimulation, participants reported that they felt more natural sensations when biomimicry was adopted, which was quantified using naturalness ratings on the visual analog scale (Figure 1E). During the stairs task, real-time biomimetic sensory feedback enabled two participants to achieve faster walking speeds and significantly improved self-reported confidence compared with either non-biomimetic somatosensory feedback or no feedback. In another functional task, the cognitive double task, two participants were asked to walk while simultaneously performing a mental task (i.e., spelling a five-letter word backward). Both participants demonstrated higher mental accuracy at the same walking speed with biomimetic stimulation compared with non-biomimetic stimulation and no feedback [68]. This study highlights that similar to findings in upper-limb neuroprosthesis users, biomimetic stimulation in lower-limb neuroprosthetics can significantly improve the functional performance and the perceived quality of tactile sensations.

## 4. Next-Generation Neuroprosthetics with Biomimetic Somatosensory Feedback

With these human trials demonstrating the significance of incorporating biomimetic somatosensory feedback in the development of high-performance bidirectional neuroprostheses, the future of neuroprosthetic technology looks increasingly promising. To further improve clinical outcomes, it is essential to not only develop advanced biomimetic neurostimulation strategies through modeling and stimulation pulse designs but also to focus on other critical components of the human–machine interface (Figure 2). These include the development of skin-like tactile sensors integrated into neuroprosthetic devices (e-skins) [82,83,84,85], advanced electronics for interfacing with sensory targets to deliver biomimetic neurostimulation [86,87], and surgical techniques for optimal access to sensory axons that transmit tactile information to the brain [88,89].

### 4.1. Electronic Skins

In 2024, Liu et al. introduced a groundbreaking electronic skin known as three-dimensionally architected electronic skin (3DAE-Skin) [90]. This design comprises three layers that structurally mimic the epidermis, dermis, and hypodermis, with thickness and elastic properties comparable with those of human skin. Additionally, its sensing components are arranged in a 3D configuration, replicating the spatial distribution of Merkel cells and Ruffini endings found in natural skin (Figure 3A) [90]. Functionally, 3DAE-Skin enables decoupled sensing of the normal force or shear force and the strain, closely resembling the specific roles of human mechanoreceptors. Its resistive sensors generate continuous signals in response to sustained stimuli, similar to the behavior of slowly adapting receptors in human skin. The researchers demonstrated that 3DAE-Skin achieves a spatial resolution for force sensing (0.117 mm) comparable with that of the human hand. Moreover, it can simultaneously detect an object’s elastic modulus and local principal curvature components through touch, similar to human tactile perception [90]. Advancements like 3DAE-Skin and other skin electronics developed by researchers at both Northwestern University [91,92,93] and Stanford University [94,95,96,97] represent a significant step forward in e-skin technology. These developments will accelerate the integration of e-skins with embedded biomimetic circuits [98] or models in commercial sensorized neuroprosthetic limbs, ultimately accelerating the clinical translation of neuroprosthetics with naturalistic somatosensory feedback [12,62].

### 4.2. Subcellular-Scale Neuroelectronic Interface

Numerous implanted electrodes have been developed and used to stimulate neurons for somatosensory feedback [24,35,37,46,102,103,104,105,106]. However, creating a stable and effective long-term neurostimulation interface remains a significant challenge [107,108]. The precise electrical stimulation of individual neurons is particularly crucial for somatosensory feedback, as mechanosensory neurons are distributed at varying depths and locations within the skin without a consistent pattern or density [60]. Non-specific stimulation risks activating multiple receptors near the implantation site, leading to distorted or inaccurate tactile perceptions. To address this issue, subcellular-scale (i.e., smaller than cellular level), high-density neuroelectronic interfaces that were tested in various animal models can potentially be a viable solution, showing promise for future applications in neuroprosthetics [99,100,109,110,111,112,113]. These interfaces, though they have not been specifically tested for providing somatosensory feedback yet, have the potential for the precise stimulation of specific groups of mechanoreceptors in response to particular tactile stimuli. In 2021, Huan et al. implanted carbon fiber electrodes into neurons of the marine mollusk Aplysia californica to evaluate intracellular stimulation in neural tissue [99]. These carbon fibers, with an extremely small diameter of approximately 8 µm, minimized damage to the cell membranes during insertion (Figure 3B). Their small size also facilitated the creation of high-density electrode arrays with pitches as small as 80–100 µm, allowing for the stimulation of multiple nearby neurons and therefore comprehensive modulation of neural activity [99]. Building on this work, Richie et al. (2024) demonstrated that sharpened carbon fiber electrodes could precisely interface with individual neurons at a subcellular level in various animal models [114]. In particular, carbon fibers were inserted into mouse retina for electrical neurostimulation. In this study, single- and multiple-cell activation of retinal ganglion cells were achieved, as evidenced by calcium response changes, using low current amplitudes (5–15 µA) [114]. In addition to the small size of the contact area between the electrode and neural tissue, it is crucial to achieve high-density stimulation to selectively interface with a large number of neurons at the implantation site [105,115]. Zhao et al. recently introduced nanoelectronic thread electrodes (NETs) that could interface with thousands of neurons over several months with a density of around 1000 neural units per cubic millimeter (Figure 3C) [100]. The high-density neural stimulation of sensory neurons, however, remains an active area of research. Nevertheless, these above findings suggest that subcellular-scale neuroelectronic interfaces hold significant potential to deliver precise somatosensory feedback in neuroprosthetics. By activating mechanoreceptors based on biomimetic strategies, these interfaces could selectively stimulate specific receptor groups to match particular tactile activities, paving the way for more accurate and naturalistic tactile feedback.

### 4.3. Regenerative Surgical Interfaces

Most previous clinical studies have utilized electrical stimulation of peripheral nerves through methods such as TENS [63,71,116,117], extraneural [19,45,79,118,119], or intraneural electrodes [46,64,120,121] to provide tactile feedback. Recently, there has been growing interest in leveraging surgical innovations to achieve more precise access to natural sensory signaling pathways in the PNS [12,88,89]. Researchers around the world are exploring the direct activation of mechanoreceptors by creating novel surgical paradigms [101,122,123,124,125]. This approach improves the specificity of tactile stimulation, as conventional nerve stimulation often activates a broad range of sensory fibers, as well as motor responses. This novel concept showed promising results in preclinical rodent models, providing more targeted and effective tactile stimulation [101,122,123]. In 2020, Svientek et al. introduced the Composite Regenerative Peripheral Nerve Interface (C-RPNI), a sensorimotor interface for individuals with limb loss [101]. This construct involves implanting a transected residual peripheral nerve between a free muscle graft and a dermal (skin) graft harvested from the plantar surface of the paw (Step 3 in Figure 2 and Figure 3D) in a rodent model. The transected nerve, which is a mixed nerve, contains both motor and sensory axons. Within this construct, motor axons preferentially reinnervate the muscle graft, while sensory axons reinnervate sensory end organs, such as mechanoreceptors in the dermal graft, as confirmed through immunostaining [101]. This surgical approach effectively isolates the sensory axons from motor axons within the peripheral nerve, creating a platform to preserve natural sensory pathways and deliver precise sensory stimulation near the reinnervated mechanoreceptors. Svientek et al. demonstrated that electrical stimulation of the dermal graft generated compound sensory action potentials (CSNAPs) recorded proximally from the nerve [101]. This shows that a dermal graft is potentially an ideal target for biomimetic neurostimulation to convey somatosensory information. In a related study, Sando et al. developed the Dermal Sensory Regenerative Peripheral Nerve Interface (DS-RPNI), which consists of a dermal graft secured around and reinnervated by a transected sensory nerve [122,126]. Both electrical and mechanical stimulation of the dermal graft produced proximal afferent neural responses, providing further evidence that both the C-RPNI and DS-RPNI constructs are viable surgical solutions for physiologically restoring sensory feedback. In 2024, Festin et al. demonstrated a modified version of the muscle–skin complex known as the biological sensorimotor interface [123]. In this construct, a mixed nerve containing both motor and sensory axons was transferred into a denervated muscle, with a glabrous skin graft placed on top of the muscle. Graded afferent responses were recorded corresponding to increasing levels of mechanical stimulation applied to the dermal graft using microfilaments. Additionally, vibration stimuli applied to the dermal graft also elicited afferent responses [123]. These animal studies above provided compelling evidence that surgically isolating motor and sensory functions within a peripheral nerve may effectively restore tactile sensation. Furthermore, incorporating mechanical stimulation through implanted motorized devices and mimicking the way natural skin processes tactile information presents a promising approach for delivering tactile feedback in future neuroprosthetic development.

In summary, e-skins, subcellular-scale neuroelectronics, and novel surgical interfaces have seen significant advancements in recent years. However, much of this progress has not advanced to the clinical trial phase yet. These ongoing efforts, combined with biomimetic encoding strategies, could significantly enhance clinical outcomes (Figure 2). The overarching goal is to improve each component of the tactile sensation loop for naturalistic tactile perception in neuroprosthetics. E-skin can detect tactile stimuli similarly to natural skin with great sensory resolution. Biomimetic encoding algorithms translate the recorded tactile information into neural spiking patterns, replicating the biological process of sensory signaling. The development of subcellular-scale neuroelectronics allows for more targeted, localized stimulation to sensory neurons, and surgical innovation further optimizes the precision to target sensory neurons. This recreated biologically relevant sensory information is then transmitted to the CNS for sensory perception, which will also guide motor control. By closing this sensorimotor loop, neuroprosthetics can achieve improved device embodiment and facilitate more intuitive use (Figure 2). The integration of biomimetic tactile sensations with other sensory inputs that are not reviewed in this paper, such as temperature [127,128,129] and proprioception [130,131,132], could bring the realization of advanced bionic limbs closer than ever. In general, these innovations are applicable to both upper- and lower-limb neuroprosthesis users with missing limbs and those with paralyzed limbs. However, regenerative surgical interfaces (Step 3 in Figure 2) in the PNS may not be a practical or effective solution for individuals with paralyzed limbs, as their impairments occur in the CNS. For such cases, interfacing with S1 directly to evoke tactile perception remains a promising avenue.

While the current development of bionic neuroprosthetics integrating biomimetic somatosensory feedback is promising, the adoption and evaluation of the long-term utility of the devices should be emphasized in future studies. Without feedback and continuous satisfaction from the users, even the most advanced bionic technology could fail. Indeed, well-coordinated team efforts are essential, from surgeons, neural engineers, and prosthetists to physical/occupational therapists, to maximize clinical outcomes with continuous support and personalized care.

## 5. Conclusions

The field of neuroprosthetics has made significant strides in recent years, particularly in the development of biomimetic somatosensory feedback systems. Clinical trials in humans showed the potential of these systems to improve the functionality and user experience of neuroprosthetic limbs. By replicating natural tactile processing, biomimetic neurostimulation approaches have demonstrated improvements in device embodiment, task performance, and intuitiveness of control. The integration of emerging advanced technologies, such as electronic skins, subcellular-scale neuroelectronic interfaces, and innovative surgical techniques, shows promise for further advancements. These developments could potentially lead to truly naturalistic tactile sensations and more effective closed-loop neuroprosthetic control. However, the successful clinical translation and broader adoption of these technologies rely on the long-term stability and effectiveness of human–machine interfaces, which have not yet been fully validated in larger populations. In particular, the users’ adoption and adherence to the neuroprosthetic devices, degree of device customization, long-term device performance, and rehabilitation need to be thoroughly evaluated in future studies. Despite these challenges, the continued development of sensorimotor neuroprosthetic limbs through biomimicry offers a promising future, with the potential to significantly improve the quality of life for individuals with limb loss or impairment.

## Figures and Tables

**Figure 1 biomimetics-10-00130-f001:**
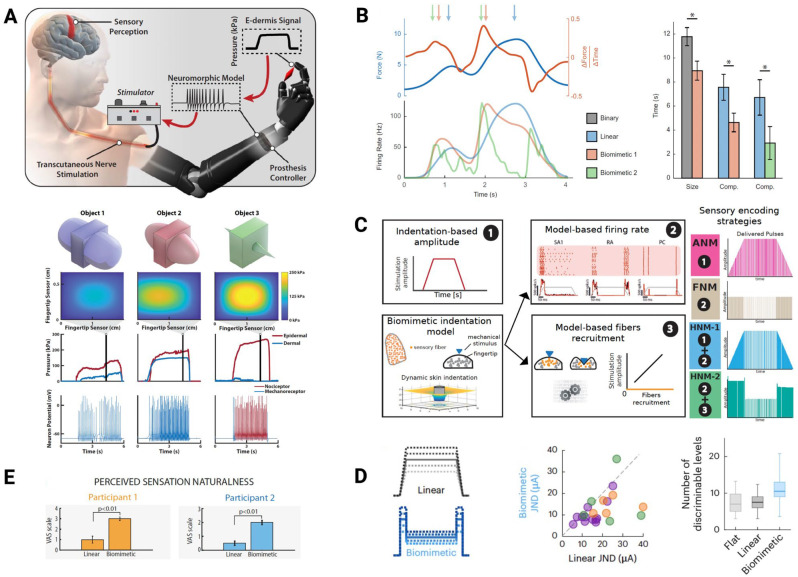
Human studies on biomimetic tactile feedback in upper-limb (**A**–**D**) and lower-limb (**E**) neuroprosthetics. (**A**) Biologically inspired prosthesis system. Neuromorphic (i.e., mimicking biological structure and function) tactile sensors combined with biomimetic (i.e., mimicking biological processes and signals) neuron models provided specific responses that corresponded to different objects. Adapted from [63]. (**B**) Biomimetic encoding strategies outperformed non-biomimetic sensory stimulation during object size and compliance discrimination tasks. * *p* < 0.05. Adapted from [64]. (**C**) Implemented and compared sensory encoding strategies, including amplitude neuromodulation (ANM), frequency neuromodulation (FNM), and hybrid neuromodulation (HNM). Adapted from [65]. (**D**) Biomimetic ICMS resulted in improved sensitivity of the electrode with reduced just-noticeable differences (JNDs) and higher resolution force feedback compared with non-biomimetic stimulation. Adapted from [47]. (**E**) Biomimetic stimulation provided more natural tactile perception (rated from 0: totally unnatural to 5: totally natural) in both participants with lower-limb amputations. Adapted from [68]. All figures were reprinted with permissions.

**Figure 2 biomimetics-10-00130-f002:**
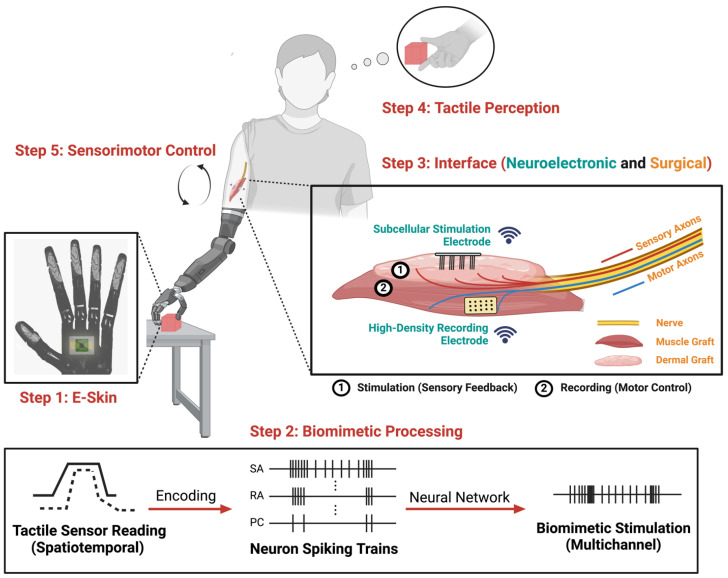
Next-generation neuroprosthetics that integrate biomimetic tactile feedback shown in steps. Flexible, electronic skin (e-skin) converts captured tactile data into biologically relevant outputs through biomimetic circuit designs. The tactile information is then processed using encoding strategies and neural networks to create multichannel biomimetic stimulation patterns. Subcellular-scale stimulation electrodes provide the precision needed to target individual neurons, selectively activating the sensory fiber populations responsible for conveying specific tactile information in response to the tactile stimuli. Meanwhile, high-density recording electrodes allow for improved decoding of motion intent. A surgical construct that biologically separates motor and sensory axons within the peripheral nerve provides optimal access to mechanoreceptors, facilitating accurate and naturalistic biomimetic tactile feedback.

**Figure 3 biomimetics-10-00130-f003:**
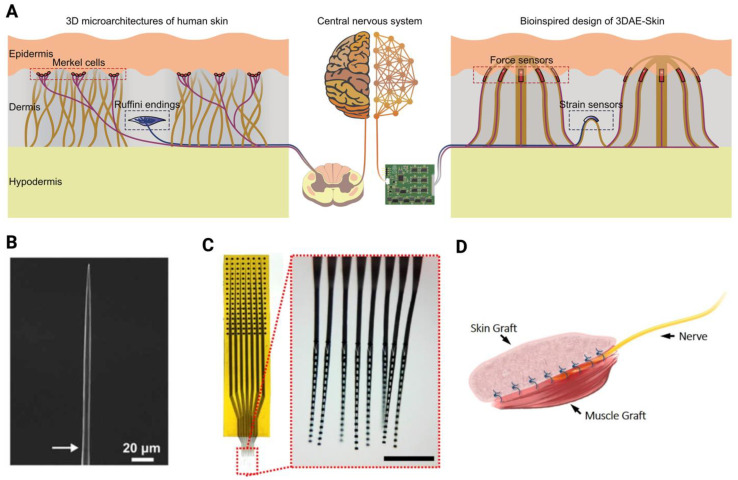
Current technologies that can potentially facilitate biomimetic tactile feedback. (**A**) E-skin. Adapted from [90]. (**B**) Sharpened subcellular electrode. Adapted from [99]. Scale bars, 500 μm. (**C**) High-density neural electrode. Adapted from [100]. (**D**) Composite Regenerative Peripheral Nerve Interface (C-RPNI) approach to access mechanoreceptors. Adapted from [101]. All figures were reprinted with permissions.

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
