# Peer review of "The Next Frontier in Neuroprosthetics: Integration of Biomimetic Somatosensory Feedback"

_biomimetics, 2025, doi:10.3390/biomimetics10030130_

Round 1

Reviewer 1 Report

Comments and Suggestions for Authors

This review covers the state of the art and future directions of sensory feedback for advanced neural prosthesis. The review is of the highest caliber, and is the product of acclaimed experts in the field. 

The manuscript is well written and the figures are exquisite. Thank you for your hard work. 

No changes requested. 

Author Response

Comments: This review covers the state of the art and future directions of sensory feedback for advanced neural prosthesis. The review is of the highest caliber, and is the product of acclaimed experts in the field. The manuscript is well written and the figures are exquisite. Thank you for your hard work. No changes requested. 

Response: Thank you very much for taking the time to review this manuscript. We are pleased to hear that you have enjoyed reading our manuscript and that you have positive feedback about the manuscript.

Reviewer 2 Report

Comments and Suggestions for Authors

This is a well-written, adequate review of the field. I have no edits.

Author Response

Comment: This is a well-written, adequate review of the field. I have no edits.

Response: Thank you very much for taking the time to review this manuscript. We appreciate the positive feedback about the manuscript.

Reviewer 3 Report

Comments and Suggestions for Authors

These review brings us closer to the latest advances in the clinical interest of in-silico neuron models to generate biomimetic somatosensory feedback provided to neuroprosthetic users through neurostimulation. Understanding that this is a narrative or historical review, as it does not present the methodological development of the database search, the inclusion criteria of the articles, or the main outcomes. In this sense, I believe it would be clarifying to state that it is a narrative review in the title. The most exposed part is that of the most recent technological strategies, which is understandable, but the inclusion of studies that have delved into other clinical variables would have been appreciated, as well as a very relevant part, which is the users' perception of the usability and adherence to the devices. 

Author Response

Comment 1: These review brings us closer to the latest advances in the clinical interest of in-silico neuron models to generate biomimetic somatosensory feedback provided to neuroprosthetic users through neurostimulation. Understanding that this is a narrative or historical review, as it does not present the methodological development of the database search, the inclusion criteria of the articles, or the main outcomes. In this sense, I believe it would be clarifying to state that it is a narrative review in the title.

Response 1: Thank you very much for taking the time to review this manuscript and pointing out the clarification we can make in the title. In the publication format, we noticed that the publisher would place "review" above our manuscript title, highlighting the purpose of the paper to the readers. We think this states that it is a review paper clearly. 

Comment 2: The most exposed part is that of the most recent technological strategies, which is understandable, but the inclusion of studies that have delved into other clinical variables would have been appreciated, as well as a very relevant part, which is the users' perception of the usability and adherence to the devices. 

Response 2: Thank you for your comment and pointing out this lack of information on the neuroprosthesis user experience in our manuscript.  We have not identified a suitable chronic clinical study that specifically evaluates user experience with using sensorimotor neuroprosthesis in the long term and therefore this manuscript focused primarily on the technology side. Nevertheless, we agree that including this in the manuscript will make the manuscript more comprehensive. Therefore, per the reviewer's suggestion, we have added this important information on Page 10 to emphasize that this is an important aspect that future clinical studies should aim to evaluate. The new added information was highlighted in red texts in the revised manuscript:

"While the current development of bionic neuroprosthetics integrating biomimetic somatosensory feedback is promising, the adoption and evaluation of the long-term utility of the devices should be emphasized in future studies. Without feedback and continuous satisfaction from the users, even the most advanced bionic technology could fail. Indeed, well-coordinated team efforts are essential, from surgeons, neural engineers, prosthetists, to physical/occupational therapists, to maximize clinical outcomes with continuous support and personalized care." 

"In particular, the users’ adoption and adherence to the neuroprosthetic devices, degree of device customization, long-term device performance, and rehabilitation need to be thoroughly evaluated in future studies” in the conclusion section.

Reviewer 4 Report

Comments and Suggestions for Authors

Biomimetic neuroprosthetics have been developed to enable tactile sensory processing, mimicking biological mechanisms to interact with the nervous system. The authors have emphasized the significance of biomimetic neuroprosthetics, offering a comprehensive review on somatosensory feedback for both upper and lower limbs. The manuscript delves into key human clinical studies that employ in-silicon neuron models to simulate biomimetic somatosensory feedback delivered to neuroprosthetic users through neurostimulation. Furthermore, it explores cutting-edge technologies such as advanced electronic skins, neural interfaces, and surgically created sensory interfaces. The manuscript is well-structured, analytically thorough, and merits consideration for publication in Biomimetics.

Author Response

Comment: Biomimetic neuroprosthetics have been developed to enable tactile sensory processing, mimicking biological mechanisms to interact with the nervous system. The authors have emphasized the significance of biomimetic neuroprosthetics, offering a comprehensive review on somatosensory feedback for both upper and lower limbs. The manuscript delves into key human clinical studies that employ in-silicon neuron models to simulate biomimetic somatosensory feedback delivered to neuroprosthetic users through neurostimulation. Furthermore, it explores cutting-edge technologies such as advanced electronic skins, neural interfaces, and surgically created sensory interfaces. The manuscript is well-structured, analytically thorough, and merits consideration for publication in Biomimetics.

Response: Thank you very much for taking the time to review this manuscript. We appreciate the positive feedback about the manuscript.